# Molecular Dynamics Simulation Study on the Interactions of Mixed Cationic/Anionic Collectors on Muscovite (001) Surface in Aqueous Solution

**DOI:** 10.3390/ma15113816

**Published:** 2022-05-27

**Authors:** Yuli Di, Ao Jiang, Haiyan Huang, Lin Deng, Dafu Zhang, Wenwei Deng, Rui Wang, Qian Luo, Shanhua Chen

**Affiliations:** 1College of Materials and Chemistry & Chemical Engineering, Chengdu University of Technology, Chengdu 610059, China; diyulidiymei@163.com (Y.D.); jiangaohb@163.com (A.J.); dling2013ls@163.com (L.D.); 15881250500@163.com (D.Z.); www.denwenweiswust@126.com (W.D.); 2School of Science, Xichang University, Xichang 615013, China; whdhcy@163.com; 3School of Biological and Chemical Engineering, Panzhihua University, Panzhihua 617000, China; 4State Key Laboratory of Comprehensive Utilization of Vanadium and Titanium, Pangang Group Research Institute Co., Ltd., Panzhihua 617000, China; 5Decision-Making Consultation Service Center of Sichuan Panzhihua, Panzhihua 617000, China; 6School of New Energy and Materials, Southwest Petroleum University, Chengdu 610500, China; wangrui20190819@163.com

**Keywords:** molecular dynamics simulation, mixed cationic and anionic collectors, muscovite, mineral flotation

## Abstract

In this study, the adsorption mechanisms of dodecylamine hydrochloride(DDAHC), sodium dodecyl sulfate (SDS), sodium dodecyl benzene sulfonate(SDBS), and their mixed anionic/cationic collectors at ten different molar ratios on a muscovite (Mcv) surface in neutral aqueous solution were assessed by molecular dynamics simulations (MDS). According to the snapshot, interaction energy, radial distribution function (RDF), and density profile between the Mcv surface and collector molecules, the individual DDAHC collector was an effective collector for the flotation of Mcv. The molar ratio of anionic/cationic collectors was determined to be an essential factor in the flotation recovery of Mcv. The DDAHC collector was involved in the adsorption of the mixed anionic/cationic collectors on the Mcv (001) surface, whereas SDS and SDBS collectors were co-adsorbed with DDAHC. The mixed cationic/anionic collector showed the best adsorption on the Mcv surface in a molar ratio of 2. Additionally, SDBS, which has one more benzene ring than SDS, was more likely to form spherical micelles with DDAHC, thus resulting in better adsorption on the Mcv surface. The results of micro-flotation experiments indicated that the DDAHC collector could improve the flotation recovery of Mcv in neutral aqueous solution, which was in agreement with MDS-derived findings. In conclusion, DDAHC alone is the optimum collector for Mcv flotation under the neutral aqueous conditions, while the mixture of DDAHC and SDBS collectors (molar ratio = 2:1) exhibits the similar flotation performance.

## 1. Introduction

Muscovite (Mcv) is a phyllosilicate mineral characterized by tetrahedral–octahedral–tetrahedral (TOT) layers with a great potential for applications in cosmetics, construction, and metallurgy industries, due to its excellent dielectric properties and high thermal stability [1,2,3]. However, Mcv is commonly associated with other gangue minerals (e.g., quartz and feldspar) in nature. Mcv is easy to cleave from the (001) surface, which is electronegative because the potassium ions on the surface are easily dissolved in water [4,5]. Dodecylamine (DDA) is a cationic collector used for the flotation separation of Mcv from gangue minerals. The adsorption behavior and flotation mechanism of DDA on Mcv have been extensively studied by molecular dynamics simulation (MDS) and experimental tests [6,7,8].

However, the mixed collectors can produce a micelle-like spherical structure more easily than the cationic or anionic surfactant collector alone due to the low micelle concentration of mixed cationic/anionic surfactant [9], thus resulting in fewer collectors and better flotation recovery of minerals. Many researchers have used mixed collectors for mineral flotation, and their results showed that the mixed collector exhibited better flotation recovery of minerals than a single collector under the same conditions [10,11]. Mixed collectors are generally divided into cationic, anionic, nonionic and amphoteric types according to their ionic type. Additionally, the synergism of mixed collectors increases with the degree of charge difference [12]. Therefore, the mixed anionic/cationic collector is the best mixed collector with the maximum flotation recovery of minerals. Yang et al. [13] studied the flotation separation of magnetite and enstatite using the mixed anionic collector sodium oleate (NaOL) and cationic collector cetyltrimethyl ammonium bromide (CTAB). The results showed that it was difficult to separate magnetite from enstatite with NaOL or CTAB collector alone, and mixed CTAB/NaOL (molar ratio = 2:1) exhibited outstanding flotation performance at pH 5.5–8.5. The mixed anionic/cationic collector was also employed for the flotation separation of kaolinite [14], feldspar [15], bastnaesite [16], hematite [17], and other minerals [18,19,20,21].

Since 1990, the mixed anionic/cationic collector has been used to separate Mcv from other minerals during the flotation process [22]. In recent years, the mixed cationic/anionic DDA/NaOL collector is one of the most widely used collectors in the flotation of Mcv minerals. Jiang et al. [23] examined the adsorption behavior of anionic NaOL, cationic DDA, and mixed DDA/NaOL collectors on the Mcv surface through dynamic contact angle measurement, atomic force microscopy analysis and flotation tests. The results demonstrated that the highest recovery (>90%) was yielded by the mixed DDA/NaOL at 2 × 10^−4^ mol/L and a ratio of 1:3. Wang et al. [24] successfully separated Mcv from quartz using the mixed DDA/NaOL collector. The results indicated that the mixed NaOL/DDA collector with 3:1 and 2:1 exhibited superior flotation separation of Mcv from quartz at pH 10 compared to DDA and NaOL alone. Xu et al. [25] conducted the surface tension analysis, contact angle measurement, adsorption analysis and flotation tests to assess the effects of solution pH, concentration of collector and ratios of cationic/anionic collector on the adsorption behavior and synergistic interaction of mixed DDA/NaOL collectors on the Mcv surface. The results indicated that the recovery of Mcv could reach a maximum value (98.45%) by the mixed DDA/NaOL with the mole ratio of 1:3 at pH 7.0. However, most of the above results were obtained by traditional flotation experiments, and there remains a lack of microscopic understanding on the adsorption mechanism of mixed DDA/NaOL at the interface. Thus, MDS has been conducted to examine the adsorption behavior of the mixed DDA/NaOL collector on the Mcv surface. Wang et al. [26] evaluated the adsorption of the mixed dodecyl amine hydrochloride (DDAHC)/NaOL collector on the Mcv surface by MDS. The results indicated that DDAHC and NaOL molecules formed a micelle structure. The DDAHC collector was absorbed on the Mcv surface via hydrogen bonding and electrostatic interactions, while the NaOL collector was interleaved and co-adsorbed with the DDAHC collector. Xu et al. [27] performed MDS to reveal the co-adsorption mechanism of the mixed DDAHC/NaOL collector on Mcv. The results of MDS demonstrated that the adsorption rates of both DDAHC and NaOL collectors were improved due to co-adsorption. The addition of NaOL decreased the electrostatic head–head repulsion between ammonium ions and the Mcv surface, while it enhanced the lateral tail–tail hydrophobic bonds. Therefore, MDS can be used as an advanced tool to explore the adsorption behavior and microscopic flotation mechanism of collector on minerals, which is generally consistent with the experimental results [28,29].

In practice, the flotation process under acidic or alkaline conditions is often employed for the recovery of fine mica, particularly mica resource in tailings [30,31,32,33]. Both the acidic and alkaline conditions require the addition of a large volume of acid or alkali to adjust the pH values, which may lead to excessive material loss and environmental pollution. Therefore, it is of great importance to evaluate the recovery of Mcv from the tailings in the neutral aqueous solution. However, limited studies have been performed on the adsorption behavior and flotation mechanism underlying the interactions between different collectors and Mcv surface in the neutral aqueous solution [34].

In this research, MDS was performed to explore the adsorption mechanisms of cationic collector DDAHC, anionic collector sodium dodecyl sulfate (SDS), anionic collector sodium dodecyl benzene sulfonate (SDBS), mixed DDAHC/SDS and DDAHC/SDBS collectors on Mcv surface under neutral conditions. Toward a better understanding of the interactions between the collectors and Mcv surface, the snapshot, interaction energy, radial distribution function (RDF) and density profile of the collectors were determined. It was found that the individual DDAHC collector was a superior collector for Mcv flotation in the neutral aqueous solution. Further micro-flotation experiments were carried out, and the findings were in good agreement with the MDS data. Our findings provide new insights into the flotation mechanisms of Mcv, which facilitate the selection of an ideal DDAHC/SDS or DDAHC/SDBS collector for the flotation recovery of Mcv fines.

## 2. Computational Details

### 2.1. Models

The crystal structures of the monoclinic C2/c 2M1 of Mcv (Figure 1) were acquired from the American Mineralogist Crystal Structure Database [35], and they were employed as the initial input structures for MDS experiments. The unit cell parameters for the monoclinic C2/c 2M1 of Mcv are: α = 90°, β = 95.78°, γ = 90°, a = 5.199 Å, b = 9.027 Å, and c = 20.106 Å. The Mcv surface was constructed along the (001) plane of structure in the middle of the interlayer space by the Cleaving Surface option in the Build tool of Materials Studio (MS). The potassium ion on the (001) surface was not included in the present study, as it could dissolve in water during actual flotation. Thus, it was deleted on the Mcv surface and subsequently added in water/collector (WC) boxes.

Firstly, the crystal structure of Mcv was optimized by the Geometry Optimization task belonging to the CASTEP module in MS version 8.0 package. The basic parameters are an energy cut-off of 340 eV, exchange–correlation potentials of general gradient approximation (GGA) + Perdew–Burke–Emzerhof (PBE) functional and *k*-point grid of 3 × 2 × 1 mesh. In addition, the spin-dependent geometrical optimization was conducted based on the convergence criteria (displacement, energy and force) of 0.03 eV/Å, and 0.001 Å, respectively. The crystal parameters of Mcv optimized by the polymer consistent force field (PCFF)–phyllosilicate force field [26,36] are α = 90°, β = 95.64°, γ = 90°, a = 5.234 Å, b = 9.087 Å, and c = 20.440 Å, with <0.5% error as a comparison of the experimental result. The optimized Mcv structure was used for subsequent modeling and MDS studies.

Secondly, the DDA cation, SDS, and SDBS anions (Figure 2) were modeled using the Visualizer tool of MS and then optimized with DMol3 module in MS. The lengths of DDA cation, SDS anion, and SDBS anion are 16.849 Å, 18.879 Å, and 21.280 Å, respectively. The atomic partial charge of DDA cation was calculated using the local density approximation–Perdew–Wang (LDA-PWC) functional, while those of SDS and SDBS anions were calculated using the PBE functional with a base set of DNP 3.5 in the GGA method. The charges of DDA, SDS, and SDBS were set to +1, −1, and −1, respectively. The convergence energy tolerance, maximum displacement, and self-consistent field (SCF) tolerance were set to 1.0 × 10^−5^ Ha, 0.005 Å, and 1 × 10^−6^ eV/atom, respectively, using the symmetric basis with a spin-unrestricted.

Thirdly, the water/collector/Mcv (WCM) system was developed using the Build Layer tool in MS based on the amorphous cell module. A supercell 5 × 3 × 1 mode of Mcv (26.1677 × 27.2596 × 18.7031 Å) was chosen with three directions of periodic boundary conditions. Simple cubic WC boxes with the same width and length on the Mcv (001) surface with 1000 water molecules and collectors were described by amorphous cell modules at 1.0 g/cm^3^. Meanwhile, the WCM system was added with K^+^, Cl^−^, and Na^+^ to ensure its electric neutrality. Table 1 lists the molar ratios of collectors and the three ions in WCM systems. Moreover, a simple cubic water box containing 200 water molecules (26.1677 × 27.2596 × 8.3875 Å) was constructed by an amorphous cell module at 1.0 g/cm^3^. Lastly, a representative WCM simulation system (Figure 3) consisting of a supercell Mcv, water box, vacuum slab, and WC boxes was constructed using the Build Layer tool. To avoid possible interactions between the two slabs, a 30 Å vacuum slab was introduced to the WCM systems. During the simulation, the water box was fixed to prevent the adsorption of the collector on the bottom surface of Mcv. In addition, the Mcv surface was also frozen to neglect the minor vibration of the minerals at ambient temperature, and the water and collectors in WC boxes were in a relaxed state.

### 2.2. Simulation Method

MDS was executed using the Forcite module in MS version 8.0 package. Firstly, a 5000-step energy minimization with the smart minimizer method was used to reduce the unreasonable contacts of WCM system. Secondly, the WCM system was conducted at the canonical ensemble under PCFF–phyllosilicate force field using the Dynamic task of the Forcite module. In addition, the temperature was maintained at 298 K using a Nose–Hoover–Langevin thermostat. The long-range electrostatic interactions were treated with the 1 × 10^−4^ kcal/mol accuracy of the Ewald summation method, and a 12.5 Å non-bond cutoff distance was set for van der Waals interactions. The MDS was conducted for 2 ns with a time step of 1 fs, and a trajectory of 2 ns was employed for data analysis. After 100 ps MDS, the relative deviations of temperature (not showed) and energy are less than 10% and 0.1%, respectively, suggesting that the systems have achieved an equilibrium state. Figure 4 showed the energies of D30, S30, SD30, D16S15, and D15SD15 simulation systems. The other systems were equilibrated after 100 ps as similar to the representative simulation systems (Figure 4).

## 3. Model Methodology

### 3.1. Snapshot of Structure

After MDS for 2 ns, a snapshot of the stable state of systems was obtained with the lowest energy. The distributions of collector, water, and other ions in the system were assessed by snapshot. In addition, the equilibrium state and adsorption behaviors of collectors on the Mcv (001) surface were also evaluated.

### 3.2. Density Profile

The density profile was used to quantitatively evaluate the properties of a simulation system as a periodic function of the positions along one axis. The changes in density profiles were attributed to varying molar ratios of collectors in the WCM system. The top surface of the Mcv was set as the zero point, and the *z*-axis was perpendicular to the c direction of the Mcv supercell. The fixed water box is not included in the analysis.

### 3.3. Radial Distribution Function (RDF)

The RDF is employed to evaluate the arrangement of water molecules within the collectors. In addition, the RDF values of water and collectors around the O atom on the Mcv (001) surface were also estimated. The RDF (*g*(*r*)) for type B around A was estimated based on Equation (1):(1)$$g_{\left(A-B\right)}\left(r\right)=\frac{1}{\rho_{B}\times 4\pi r^{2}}\times \frac{dN_{A-B}}{dr}$$
where *ρ_B_* denotes the density of type *B*, *r* represents the distance between *B* and *A*, and *dN_A−B_* indicates the average number of type B from *r* to *r + dr* compared to the reference type *A*. The fixed water box is not included in the analysis.

### 3.4. Interaction Energy

The interaction energy was used to quantify the interactions between collectors with Mcv, which can be expressed by Equation (2):(2)$$E_{inter}=\frac{E_{total}-(E_{col}+E_{mus)}}{n}$$
where *E_total_* (kcal/mol) is total energy (Figure 5b); *E_col_* and *E_mus_* (kcal/mol) are the energies of collectors (Figure 5c) and Mcv (Figure 5d), respectively; and *n* denotes the number of collectors. For clarity, the calculation method is demonstrated in Figure 5.

## 4. Micro-Flotation Experiments

The Mcv samples were supplied by Malipo County (Wenshan Prefecture, Yunnan province, China). The samples were ground in a porcelain mill with an agate ball. The obtained products were subjected to dry screening to achieve the particle size of 38–74 μm for subsequent micro-flotation experiments. An X-ray Fluorescence (XRF) Spectrometer was used to assess the chemical compositions of Mcv samples (Table 2).

The cationic DDA surfactant, anionic SDS and SDBS surfactants with an analytical grade were obtained from Shanghai Macklin Biochemical and Chengdu Ouen Ruisi Chemical Reagent, respectively, which were used as collectors. Equimolar levels of DDA and hydrochloric acid were mixed to prepare dodecylamine hydrochloride (DDAHC). The DDAHC/SDS and DDAHC/SDBS collectors were prepared by mixing DDAHC and SDS or SDBS under the same concentration, respectively. To avoid precipitation, the mixture was prepared freshly. Distilled water was employed in all tests.

The micro-flotation experiments were performed using an XFG5-35 flotation machine (60 mL cell, 1600 rpm spindle speed). To prepare a mineral suspension, 2.0 g Mcv was added and then filled with distilled water to 50 mL. The surfactant reagent was added into the cell and conditioned for 2 min. After flotation for 5 min, filtration and drying, the flotation products and tailings were separately weighed. The flotation recovery was then calculated. To determine the average recovery value, each flotation experiment was repeated 3 times under the same conditions. The following equation was used to calculate flotation recovery:(3)$$\varphi =\frac{m_{1}}{m_{1}+m_{2}}\times 100\%$$
where *φ* (%) denotes the flotation recovery; *m*_1_ and *m*_2_ (g) represent the values of the flotation products and tailings, respectively.

## 5. Results and Discussion

### 5.1. Adsorption Behavior of Single Collector on Mcv (001) Surface

#### 5.1.1. DDAHC Collector

The Mcv (001) surface is negatively charged owing to the isomorphous substitution of Al^3+^ for Si^4+^. Thus, DDAHC can be absorbed on the surface via electrostatic interactions. As shown in Figure 6, the polar head group (-NH_3_) of DDAHC is adsorbed onto the bottom and top surfaces of Mcv, and the alkyl chains (ACs) of DDAHC were intertwined with each other and formed a hydrophobic membrane, thus generating a hydrophobic state that was the same as the MDS results by Wang et al. [26,37,38]. Some DDAHC molecules are removed from the Mcv surface, suggesting that the adsorption of the DDAHC collector reaches a saturation degree. The three H atoms of -NH_3_ were bonded to the O atom of the Mcv surface, leading to four or five hydrogen bonds (Figure 6b). Additionally, the -NH_3_ of DDAHC and K^+^ in water were both located on the cavities of [Si_5_Al_1_O_6_] and [Si_4_Al_2_O_5_] (Figure 6b,c), but the -NH_3_ was located near the Mcv surface, demonstrating that DDAHC is likely to be adsorbed on the electronegative Mcv (001) surface compared with K ions.

As displayed in Figure 7, two peaks were observed at 0.546 and 1.059 Å, which corresponded to the N atom of the DDAHC head group (-NH_3_), suggesting that a DDAHC collector is adsorbed onto the surfaces of Mcv. The total density of the adsorptive N atom (-NH_3_) in the D30 system was 0.699 (0.381 + 0.318) g/cm^3^. Based on the density profiles of C6 and C12, the AC values of the DDAHC collector were intertwined with each other and formed aggregate structures at a great distance from the Mcv surface. These results are in agreement with the MDS data presented in Figure 6. The RDF values of the D30 system are demonstrated in Figure 8. The water and DDAHC collector are located near the O atom of the Mcv surface. As displayed in Figure 8a, the first peak positions of RDF indicated that the distances from the H atom of water, O atom of water, N atom of DDAHC, C6 atom of DDAHC, and C12 atom of DDAHC to the O atom of the Mcv surface were in the following order: H atom of water < N atom of DDAHC < O atom of water < C6 atom of DDAHC = C12 atom of DDAHC. The positively charged H atoms of the DDAHC head group (-NH_3_) and water could be strongly attracted by the electronegative Mcv surface, thus leading to DDAHC adsorption. The distances from the peak positions of N and C12 atoms in DDAHC to those of O and H atoms in water were subsequently compared (Figure 8b). It was found that the hydrophyllic head group (-NH_3_) of DDAHC was closer to water than the hydrophobic AC of DDAHC.

#### 5.1.2. SDS Collector

As demonstrated in Figure 9, the adsorption state of the SDS collector on the Mcv surface was distinct from that of DDAHC, indicating that the SDS collector is not obviously adsorbed on the Mcv surface. Most of the SDS molecules formed an aggregate structure and were located far from the Mcv surface (Figure 10). However, both Na^+^ and K^+^ ions were absorbed on the six-membered cavity of the Mcv surface. The distances from the Na^+^ or K^+^ ion to the Mcv surface lay within the range of 1.316–1.924 Å, but the Na ions were nearer to the Mcv surface than K ions. As displayed in Figure 10, the nearest C12 atom peak of SDS was at 21.873 Å from the Mcv surface. Based on the density profiles of C7 and C12 of SDS, the carbon atoms of the SDS collector were intertwined with each other and formed aggregate structures at a great distance from the Mcv surface. These results are in line with the MDS data revealed in Figure 9. As a result, the SDS collector is not to be adsorbed on the Mcv surface, leading to its poor flotation performance.

The RDF values of the S30 system are presented in Figure 11. As demonstrated in Figure 11a, the first peak positions of RDF indicated that the distances from the H atom of water, O atom of water, S atom of SDS, C7 atom of SDS, and C12 atom of SDS to the O atom of the Mcv surface were in the following order: H atom of water < O atom of water < C7 atom of SDS = C12 atom of SDS < O atom of SDS < S atom of SDS. The positively charged H atom of water was attracted to the Mcv (001) surface, while the negatively charged SDS head group (-SO_3_) was repelled from the Mcv surface, leading to no SDS adsorption. The distances from the peak positions of S and C12 atoms in SDS to those of O and H atoms in water were compared (Figure 11b). It was observed that the hydrophyllic head group (-SO_3_) of SDS was closer to water than the hydrophobic AC of SDS.

#### 5.1.3. SDBS Collector

As displayed in Figure 12, the adsorption state of the SDBS collector on the Mcv surface was different compared with DDAHC and SDS, implying that the SDBS collector is not obviously adsorbed on the Mcv surface. Most of the SDBS molecules formed aggregate structures and were located far from the Mcv surface (Figure 13), similarly toSDS molecules. However, similarly to the S30 system, both Na^+^ and K^+^ ions of the SD30 system were absorbed on the six-membered cavity of the Mcv surface. The distances from Na^+^ and K^+^ ions to the Mcv surface lay within the range of 1.362–2.015 Å, but Na ions were nearer to the Mcv surface than K ions. As demonstrated in Figure 13, the nearest C12 atom peak of SDBS was at 21.848 Å from the Mcv surface. Based on the density profiles of C5 and C12 of SDBS, the carbon atoms of the SDBS collector were intertwined with each other and formed aggregate structures at a great distance from the Mcv surface. These results are consistent with the MDS data shown in Figure 12. As a consequence, no SDBS collector is expected to be adsorbed on the Mcv surface, leading to its poor flotation performance.

The RDF values of the SD30 system are presented in Figure 14. The water was located near the O atom of the Mcv surface. As shown in Figure 14a, the first peak positions of RDF indicated that the distances from the H atom of water, O atom of water, S atom of SDBS, C5 atom of SDBS, and C12 atom of SDBS to the O atom of Mcv surface were in the following order: H atom of water < O atom of water < C12 atom of SDBS < C5 atom of SDBS < S atom of SDBS. It can be seen that the SDBS collectors are farther from the mineral surface than water molecules, indicating that SDBS collectors are almost separated from the muscovite surface [39]. The distances from the peak positions of S and C12 atoms in SDBS to those of O and H atoms in water were compared (Figure 14b). It was noted that the hydrophilic head group (-SO_3_) of SDBS was closer to water than the hydrophobic AC of SDBS, which is consistent with its physical properties.

In summary, the MDS of the adsorption mechanism of Mcv with a single collector showed that the H atoms of the head group (-NH_3_) of DDAHC could form hydrogen bonds with the O atoms of Mcv (001) surface, thus resulting in DDAHC adsorption on Mcv. For the other two anionic collectors, they are located far from the Mcv surface, leading to no adsorption on the Mcv surface. The cationic collector DDAHC had an obvious adsorption behavior on Mcv, indicating that it exhibited good flotation performance toward Mcv. The other two anionic collectors showed poor flotation performance of Mcv because of no obvious adsorption behavior.

### 5.2. Adsorption Behavior of Mixed Collectors on Mcv (001) Surface

#### 5.2.1. DDAHC/SDS Collectors

Figure 15 shows the equilibrium configuration of mixed DDAHC/SDS collectors at different molar ratios on the Mcv (001) surface in the aqueous solution. The mixed DDAHC/SDS collectors on the Mcv surface had a distinct aggregate morphology compared with the DDAHC or SDS collector alone. Regardless of the molar ratios, the head group (-NH_3_) of DDAHC could be adsorbed on the Mcv (001) surfaces. The peaks of N atoms in the DDAHC head groups were located at the distances of 0.552–1.062 Å from the Mcv (001) surfaces (Figure 16), suggesting that the DDAHC collector is greatly adsorbed on the Mcv surface. The density peaks of N atoms in the D8S23, D10S20, D16S15, D20S10, and D23S8 systems were compared. It was found that the DDAHC molecules of the D20S10 system interacted more strongly with the Mcv surface compared to other systems: D20S10 (0.510 g/cm^3^) > D23S8 (0.382 g/cm^3^) > D10S20 (0.322 g/cm^3^) > D16S15 (0.274 g/cm^3^) > D8S23 (0.130 g/cm^3^). The highest total density of adsorptive N atoms of the DDAHC head group (-NH_3_) in the D20S10 system was 0.510 g/cm^3^, which is still smaller than that of the D30 system (0.699 g/cm^3^). On the basis of the research of Bai Yang et al. [29], the mixed DDAH/SDS collector has much lower zeta potential than DDAH or SDS alone because the positive charge of DDA cations were neutralized by the SDS anions, indicating that the mixed DDAH/SDS has a synergistic effect on the adsorption of minerals. However, due to the absence of H^+^ or OH^−^ ions under neutral conditions, the synergistic effect of mixed collectors would be weakened, resulting in poor adsorption on muscovite than DDAHC alone. Therefore, the density of adsorptive N atoms of DDAHC alone on muscovite is higher than the mixed collectors. Comparing the density distribution of DDAHC and SDS, the DDAHC molecules are much closer to the surface of Mcv than SDS molecules, indicating that DDAHC has better adsorption capacity than SDS. For all the mixed DDAHC/SDS collector systems, the nearest S atom peak of the SDS head group (-SO_3_) was located at 10.054 Å from the Mcv surface, implying that SDS is not adsorbed on the Mcv surface. However, the SDS collector was hinged with the DDA collector, thus improving SDS co-adsorption on the Mcv surface. With the increase in DDAHC, the amount of DDAHC adsorbed on the Mcv surface was also increased. The D20S10 system had the best adsorption of DDAHC than other systems. Therefore, it is speculated that DDAHC plays a major role in regulating the interactions between the mixed DDAHC/SDS collector and Mcv surface.

The representative RDF values of the mixed DDAHC/SDS systems are shown in Figure 17 and Figure 18. As displayed in Figure 17, the first peak positions of RDF demonstrated that the distances from the H atom of water, N atom of DDA, O atom of water and S atom of SDS to the O atom of Mcv surface were in the following order: H atom of water < N atom of DDA < O atom of water < S atom of SDS. The positively charged H atoms of the DDAHC head group (-NH_3_) and water could be attracted by the electronegative Mcv surface, leading to DDAHC adsorption. In addition, the H atoms of water also can form hydrogen bonding interaction to the O atoms of the Mcv surface and head groups of DDAHC [8,24,28], resulting in the formation of a water film between the adsorbed DDAHC collector and the Mcv surface. The distances from the peak positions of S and C12 atoms in SDS to those of N and C12 atoms in DDAHC were compared (Figure 18). The results showed that the S atom of SDS was positively charged and was nearer to the negatively charged N atom of DDAHC than the C12 atom, while the C12 atom of SDS was nearer to that of DDAHC than the N atom. Additionally, the head group (-NH_3_) of DDAHC was inclined to intertwine with the head group (-SO_3_) of SDS rather than that of AC.

#### 5.2.2. DDAHC/SDBS Collectors

Figure 19 shows the equilibrium configuration of mixed DDAHC/SDBS collectors at varying molar ratios on the Mcv (001) surface in the aqueous solution. The mixed DDAHC/SDBS collectors on the Mcv surface had a distinct aggregate morphology compared with the DDAHC or SDBS collector alone. Regardless of the molar ratios, the head group (-NH_3_) of DDAHC could be adsorbed on the Mcv surfaces. The peak positions of N atoms in the DDAHC head group were at the distances of 0.657–1.160 Å from the Mcv surfaces (Figure 20), implying that the DDAHC collector is greatly adsorbed on the Mcv surface. The density peaks of N atoms in the D8SD23, D10SD20, D15SD15, D20SD10 and D23SD8 systems were compared. It was observed that the DDAHC molecule of the D20SD10 system interacted more strongly with the Mcv surface compared to the other systems: D20SD10 (0.517 g/cm^3^) > D23SD8 (0.258 g/cm^3^) > D15SD15 (0.254 g/cm^3^) > D8SD23 (0.130 g/cm^3^) > D10SD20 (0.129 g/cm^3^). The highest total density of adsorptive N atoms of the DDAHC head group (-NH_3_) in the D20SD10 system was 0.517 g/cm^3^, which is still smaller than that of the D30 system (0.699 g/cm^3^). Learning from the research of Vidyadhar et al. [10] and Tian et al. [21], different charge collectors can reduce the repulsion force of adsorption layer and promote the formation of semi-micelle adsorption. We believe that under the action of the head group of SDBS in acidic or alkaline conditions, the electrostatic repulsion between DDA cations is obviously weakened. Moreover, the adsorption performance of the mixed cationic/anionic collectors on the muscovite surface is theoretically improved, thus increasing the flotation recovery of muscovite. It is obvious that this phenomenon is not suitable for neutral conditions, because the pH value of solution will lead to changes in electrostatic and hydrogen bonding interactions, resulting in different flotation results, which is consistent with the flotation experiments as shown in Section 5.4. For all the mixed DDAHC/SDBS collector systems, the nearest S atom peak of the SDBS head group (-SO_3_) was located at 7.865 Å from the Mcv (001) surface, suggesting that SDBS is not adsorbed on the Mcv surface. Consistent with the situation of a mixed DDAHC/SDS collector, DDAHC was still the main adsorption on the Mcv surface, while SDBS and DDAHC hinged together to achieve co-adsorption. The cationic DDAHC and anionic SDBS collectors hinged with each other into spherical or elliptic micelles, which was better than mixed DDAHC/SDS collectors. SDBS has one more benzene ring than SDS, which can form a micelle hinged with DDAHC relatively well. With the increase in DDAHC content, it is evident that the amount of collector floating on the water surface decreases, indicating that there are more anionic collectors that hinge with DDAHC.

The representative RDF values of the mixed DDAHC/SDBS systems are shown in Figure 21 and Figure 22. As shown in Figure 21, the first peak positions of RDF indicated that the distances from the H atom of water, O atom of water, N atom of DDA, and S atom of SDBS to the O atom of Mcv surface were in the following order: H atom of water < N atom of DDA < O atom of water < S atom of SDBS. The positively charged H atoms of the DDAHC head group (-NH_3_) and water could be attracted by the electronegative Mcv surface, leading to DDAHC absorption. There is a water film between the adsorbed DDAHC and muscovite surface regardless of SDS or SDBS anionic collector, which indicated that the DDAHC plays a major role in the adsorption of Mcv under mixed cationic/anionic collectors [26,34]. The distances from the peak positions of S and C12 atoms in SDBS to those of N and C12 atoms in DDAHC were compared (Figure 22). The results demonstrated that the S atom of SDBS was positively charged and was nearer to the negatively charged N atom of DDAHC than the C12 atom, while the C12 atom of SDBS was closer to the C12 atom of DDAHC than the N atom. Thus, the head group (-NH_3_) of DDAHC is inclined to intertwine with the head group (-SO_3_) of SDBS more than AC, similarly to the mixed DDAHC/SDS collectors.

### 5.3. Interaction Energy

The interaction energy of the collectors on Mcv in all systems is listed in Table 3. The negative and positive values of the interaction energy *E_inter_* indicate an effective and no adsorption between the collectors and Mcv surface, respectively. (SDS), anionic collector. 

The positive interaction energy between the SDBS or SDS and Mcv demonstrated that the SDS or SDBS had no absorbability on Mcv, while the negative interaction energy of the mixed collector systems indicated that the mixed collectors had effective adsorption on Mcv. The interaction energy of DDAHC in the D30 system was −851.51 kcal/mol, which was highest among all the systems. The interaction energy of the mixed DDAHC/SDS collector systems was lower than that of DDAHC but higher than that of SDS, following the sequence: D20S10 < D23S8 < D16S15 < D10S20 < D8S23. Moreover, the mixed DDAHC/SDBS systems had more negative values than SDBS but fewer than DDAHC, following the sequence: D20SD10 < D23SD8 < D15SD15 < D10SD20 < D8SD23. The interaction energy of mixed DDAHC/SDBS systems is more negative than that of mixed DDAHC/SDS systems, indicating that the mixed DDAHC/SDBS systems have better flotation performance on Mcv than the mixed DDAHC/SDS collector. Above all, in the mixed collector system, the interaction energy of the cationic/anionic collector in a molar ratio of 2 is the most negative, indicating that it had the best flotation performance toward Mcv. Furthermore, the interaction energies of mixed DDAHC/SDS and DDAHC/SDBS collectors in a molar ratio of 2 were −590.17 and −665.19 kcal/mol, respectively. According to the literature survey, the interaction energy of different mineral–collector systems is obviously different, which is listed in Table 4. With the exception of SDS and SDBS collectors, the interaction energy in our work ranged from −205.65 to −851.53 kcal/mol, indicating that the interaction energy between muscovite and collectors is relatively large. Combined with the snapshot results, the SDBS collector was more likely to form spherical micelles with DDAHC due to the addition of a benzene ring structure and had more negative interaction energy on Mcv. However, compared with the mixed collector, the DDAHC alone was much more effective toward Mcv in neutral aqueous solution.

### 5.4. Flotation Recovery of Mcv with Cationic, Anionic and Mixed Collectors

Micro-flotation experiments were conducted to verify the MDS results. Figure 23 shows the flotation recovery of Mcv with DDAHC, SDS, SDBS, mixed DDAHC/SDS and DDAHC/SDBS collectors. Under neutral conditions, the flotation performance of the mixed DDAHC/SDS or DDAHC/SDBS collectors at molar ratios of 1:1, 2:1, and 3:1 was greater than that of SDS or SDBS but lower than that of DDAHC alone. The flotation recovery of Mcv with mixed DDAHC/SDS or DDAHC/SDBS collectors at molar ratios of 2:1 and 3:1 was increased compared to those at 1:1, 1:2, and 1:3. The flotation recovery rates of Mcv with mixed DDAHC/SDS collectors at 2 × 10^−2^ mol/L and molar ratios of 1:3, 1:2, 1:1, 2:1, and 3:1 were 9.65 ± 1.12, 14.63 ± 1.74, 35.45 ± 2.03, 64.29 ± 3.08, and 60.8 ± 2.51%, respectively. Moreover, the flotation recovery rates of Mcv with mixed DDAHC/SDBS collectors at 9 × 10^−3^ mol/L and molar ratios of 1:3, 1:2, 1:1, 2:1, and 3:1 were 7.09 ± 2.72, 23.43 ± 1.87, 28.42 ± 1.72, 81.66 ± 2.45, and 75.19 ± 2.16%, respectively. In addition, at the same concentration of collector, the flotation recovery of the mixed DDAH/SDBS collector is much higher than that of the mixed DDAH/SDS collector, which is consistent with the results of interaction energy. However, under neutral conditions at 7 × 10^−3^ mol/L, DDAHC alone (D30) exhibited a maximum flotation recovery of 81.79 ± 2.67% on Mcv. Hence, the MDS results of D20S10, D20SD10, and D30 on Mcv are in good agreement with the micro-floatation experimental data at 7 × 10^−3^ mol/L. According to the flotation results, under neutral conditions, the flotation recovery of muscovite using the DDAHC collector alone is similar to that of the mixed collector DDA/SDBS in a molar ratio of 2:1. However, the single collector is more convenient than the mixed collector in practical application; it can be concluded that DDAHC is the optimum collector for the flotation of muscovite in neutral aqueous solution.

## 6. Conclusions

In the present work, the effects of DDAHC, SDS, SDBS, and their mixture with ten molar ratios on the adsorption behavior of the Mcv surface under neutral conditions were investigated by the MDS studies. The cationic collector DDAHC could be adsorbed on the Mcv surface, owing to the hydrogen bond and electrostatic interactions between the collectors and the Mcv surface. However, the anionic collectors SDS and SDBS could not be adsorbed on the Mcv surface. The mixed DDAHC/SDS and DDAHC/SDBS collectors with the molar ratios of 1:1, 1:2, and 1:3 exhibited a poor adsorption on Mcv, while those with the molar ratios of 2:1 and 3:1 demonstrated an effective adsorption on Mcv. DDAHC plays an important role in regulating the interactions between the mixed collectors and Mcv surface. The anionic and cationic collectors hinged with each other, resulting in the co-adsorption of SDS or SDBS on the Mcv surface. The results of MDS and micro-flotation experiments in the presence of DDAHC, SDS, SDBS, DDAHC/SDS, and DDAHC/SDBS collectors on the Mcv surface demonstrated that DDAHC alone (7 × 10^−3^ mol/L) had the best flotation performance (81.79%) toward Mcv under neutral aqueous conditions. Additionally, the mixed DDAHC/SDBS collector (81.66%) exhibited better flotation recovery than DDAHC/SDS (64.29%) on Mcv, which was probably due to the fact that SDBS has one more benzene ring.

## Figures and Tables

**Figure 1 materials-15-03816-f001:**
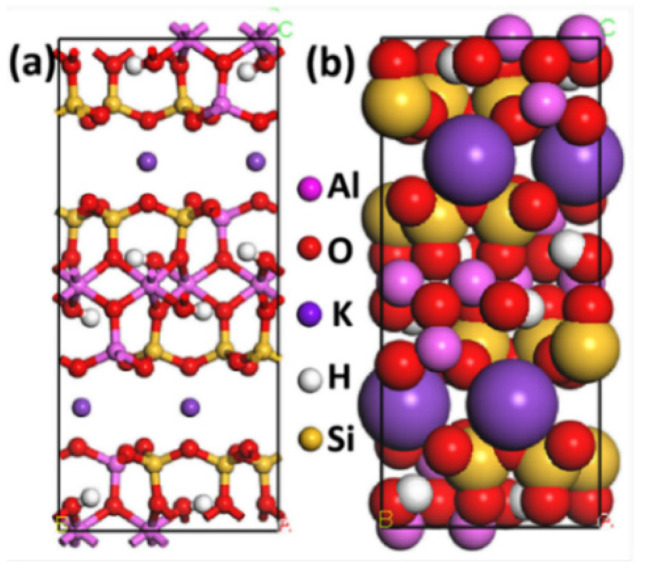
(**a**) The ball and stick model and (**b**) CPK model of the Mcv cell. The colors of yellow, pink, red, and purple represent silicon, aluminum, oxygen, and potassium, respectively. The model of Mcv was used as the initial input structure for our MDS, which is used in our previous work [34].

**Figure 2 materials-15-03816-f002:**
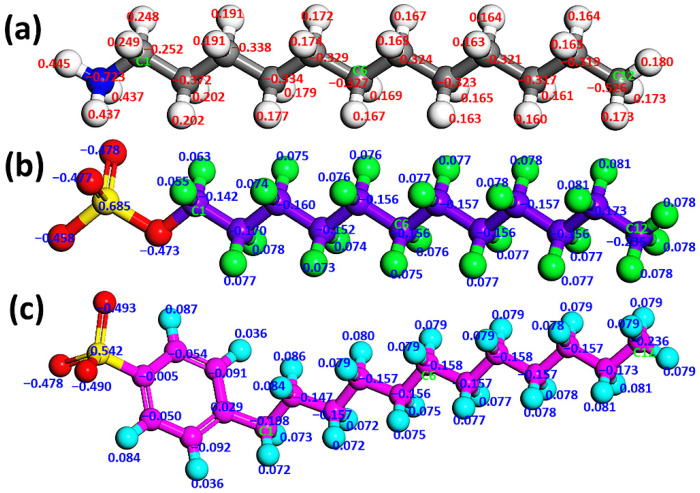
Structures and atom numbering of (**a**) DDA cation, (**b**) SDS anion, and (**c**) SDBS anion. In (**a**), the colors gray, blue, and white represent carbon, nitrogen, and hydrogen atoms, respectively. In (**b**), the colors red, green, yellow, and purple represent oxygen, hydrogen, sulfur, and carbon atoms, respectively. In (**c**), the colors red, pink, yellow, and light blue represent oxygen, carbon, sulfur, and hydrogen atoms, respectively. Corresponding carbon atom numbers are given according to the order in which the carbon atoms are arranged, where C1, C6 and C12 represent the first, sixth and 12th carbon atoms respectively.

**Figure 3 materials-15-03816-f003:**
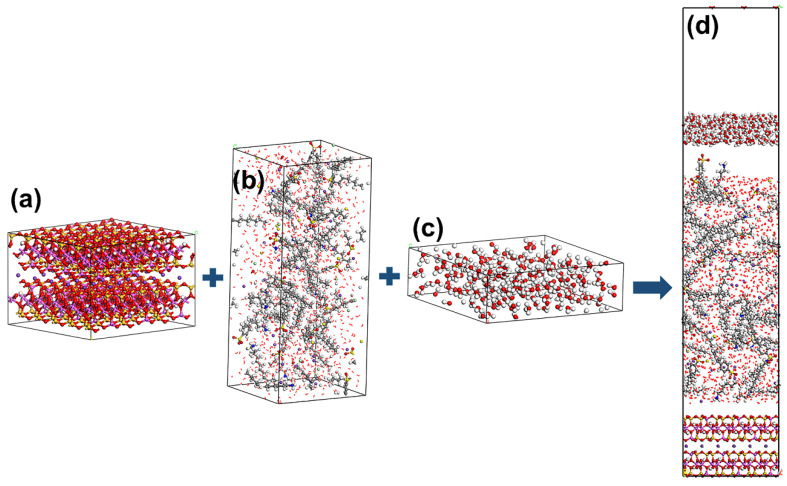
Schematic diagram of the modeling process: (**a**) supercell 5 × 3 × 1 mode of Mcv, (**b**) WC box, (**c**) water box, and (**d**) the WCM simulation system. The different colors represent different atoms, which are the same as in Figure 1 and Figure 2.

**Figure 4 materials-15-03816-f004:**
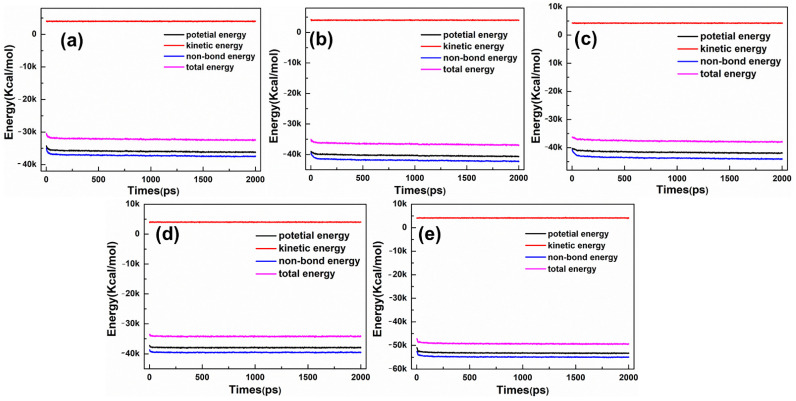
The energy profiles of the representative simulation systems: (**a**) D30, (**b**) S30, (**c**) SD30, (**d**) D16S15, and (**e**) D15SD15.

**Figure 5 materials-15-03816-f005:**
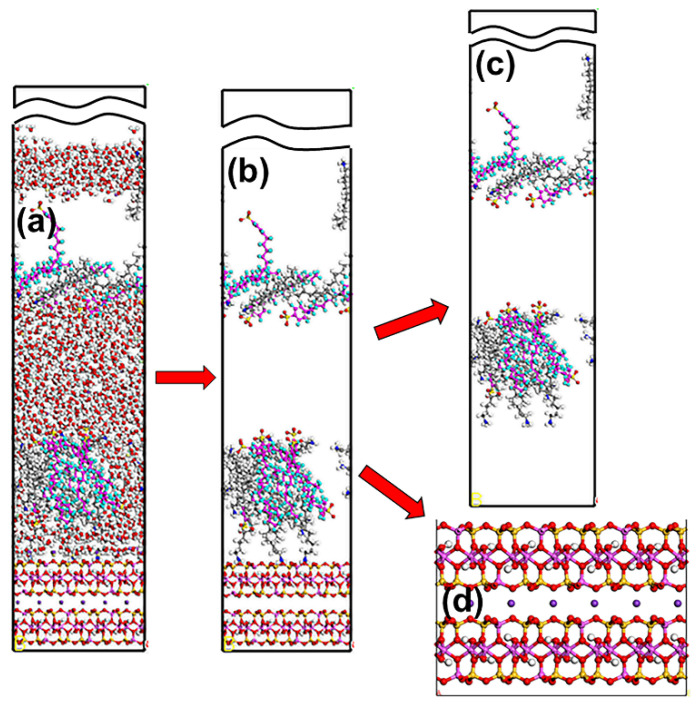
Schematic diagram for the calculation method of the interaction energy: (**a**)original structure; (**b**) calculated total energy of structure; (**c**) calculated collectors energy of structure and (**d**) calculated muscovite energy of structure. The break lines indicate regions of the vacuum slab. The different colors represent different atoms, which are the same as in Figure 1 and Figure 2.

**Figure 6 materials-15-03816-f006:**
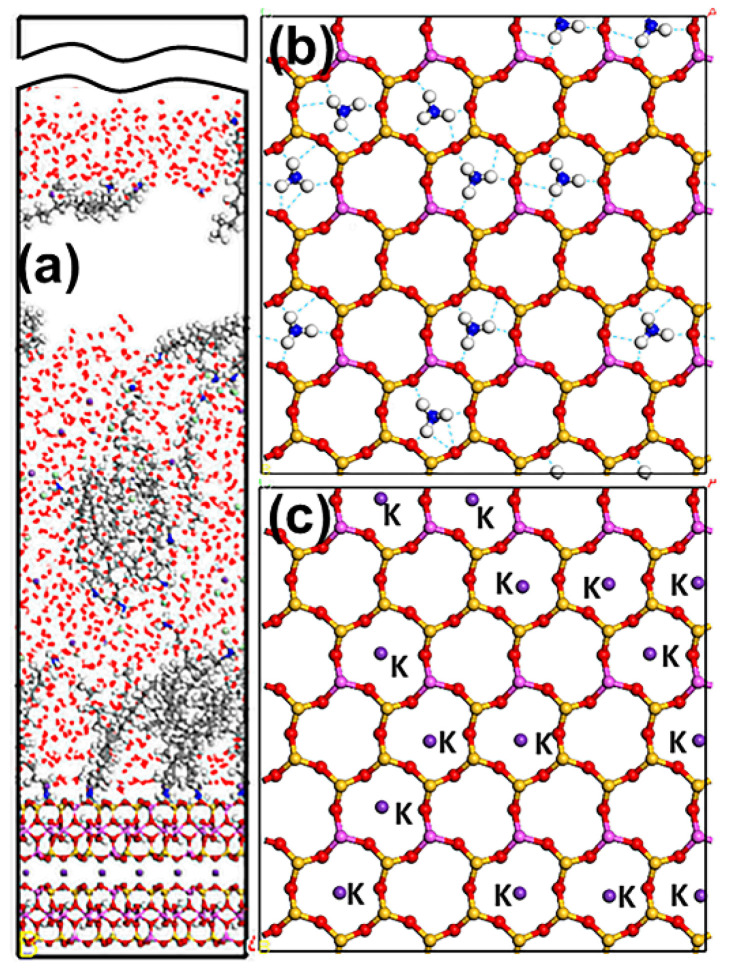
Snapshots of the MDS of 30 DDAHC molecules forming on the Mcv (001) surface, showing (**a**) the equilibrium structure of the D30 system, (**b**) overview of the hydrogen bonds (dashed line) between the head group of DDAHC and Mcv surface, and (**c**) overview of the K ions located above the cavities of the Mcv surface. For clarity, only the water molecules are shown in the line model. The different colors represent different atoms, which are the same as in Figure 1 and Figure 2. The break lines indicate regions of the vacuum slab.

**Figure 7 materials-15-03816-f007:**
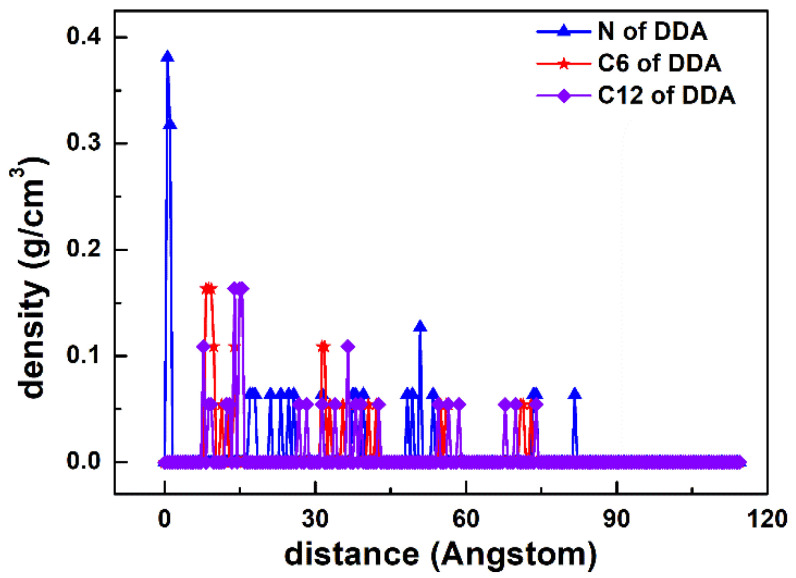
The density distribution profiles along the z-direction of N, C6, and C12 of DDA.

**Figure 8 materials-15-03816-f008:**
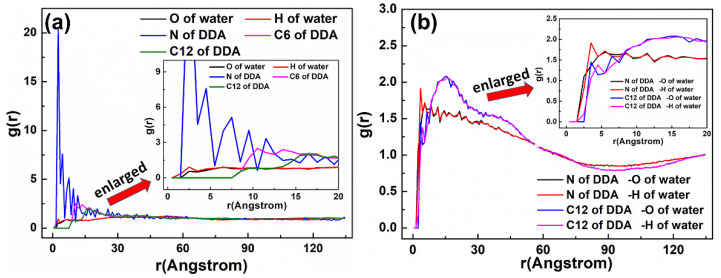
RDF g(r) between the DDAHC collector and (**a**) the Mcv (001) surface and (**b**) water molecules, respectively.

**Figure 9 materials-15-03816-f009:**
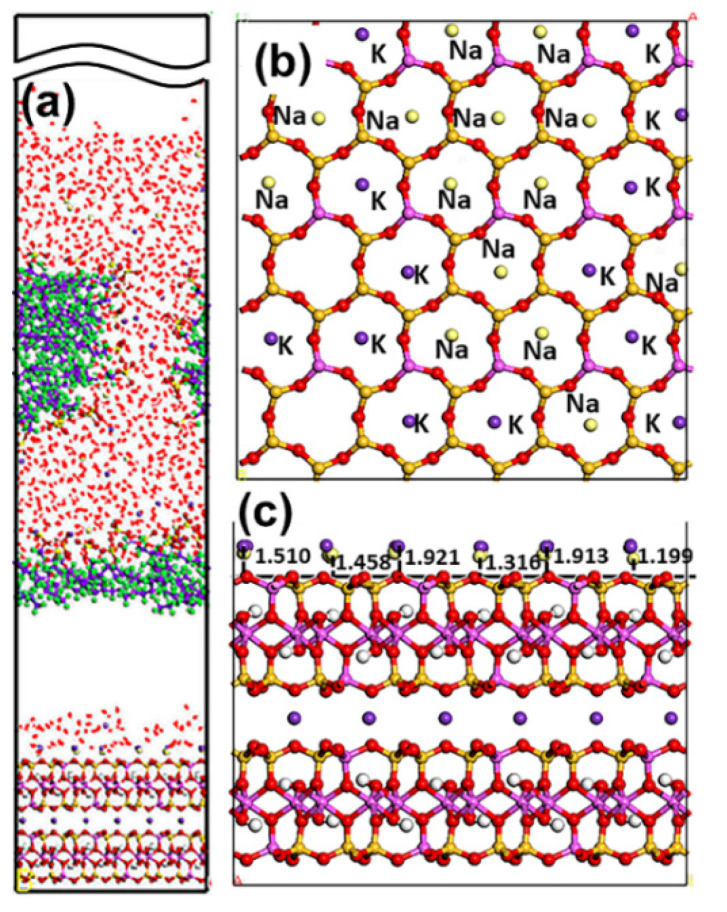
Snapshots of the MDS of 30 SDS molecules forming on the muscovite (001) surface, showing (**a**) the equilibrium structure of the S30 system, (**b**) over and (**c**) side view of the K and Na ions located above the cavities of the Mcv surface. For clarity, only the water molecules are shown in the line model. The different colors represent different atoms, which are the same as in Figure 1 and Figure 2. In addition, the colors light green and light yellow represent chloride and sodium, respectively. The break lines indicate regions of the vacuum slab.

**Figure 10 materials-15-03816-f010:**
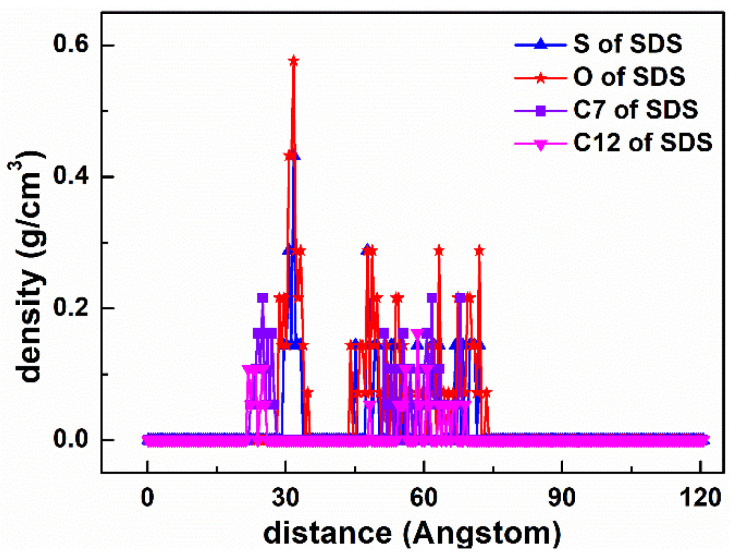
The density distribution profiles of S, O, C7, and C12 of SDS along the z-direction.

**Figure 11 materials-15-03816-f011:**
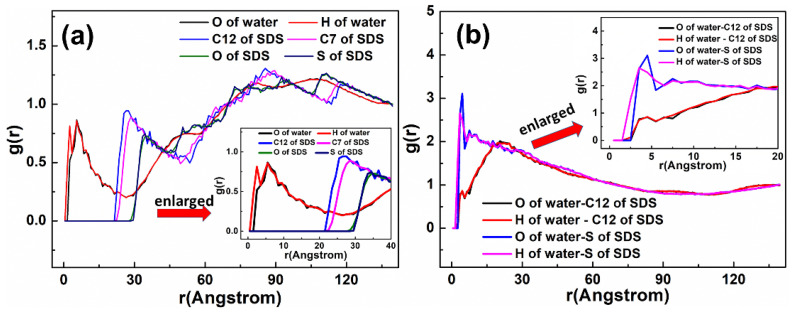
RDF g(r) between the SDS collector and (**a**) the Mcv (001) surface and (**b**) water molecules, respectively.

**Figure 12 materials-15-03816-f012:**
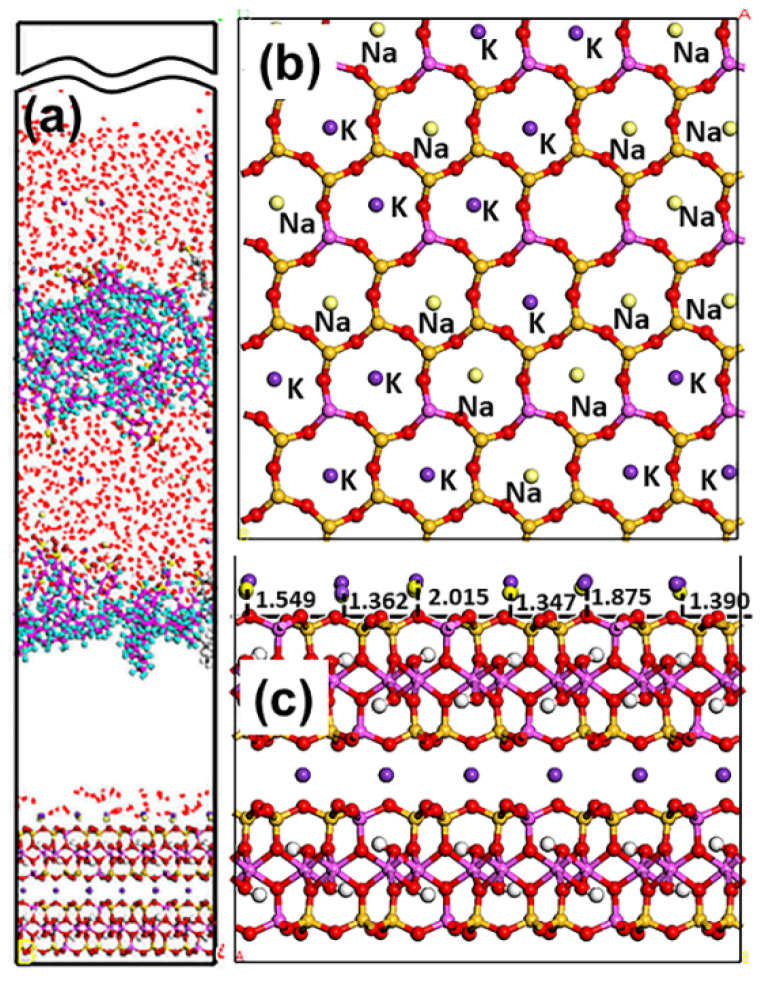
Snapshots of the MDS of 30 SDBS molecules forming on the muscovite (001) surface, showing (**a**) the equilibrium structure of the SD30 system, (**b**) overview and (**c**) side view of the K and Na ions located above the cavities of the muscovite surface. For clarity, only the water molecules are shown in the line model. The different colors represent different atoms, which are the same as in Figure 1 and Figure 2. In addition, the colors light green and light yellow represent chloride and sodium, respectively. The break lines indicate regions of the vacuum slab.

**Figure 13 materials-15-03816-f013:**
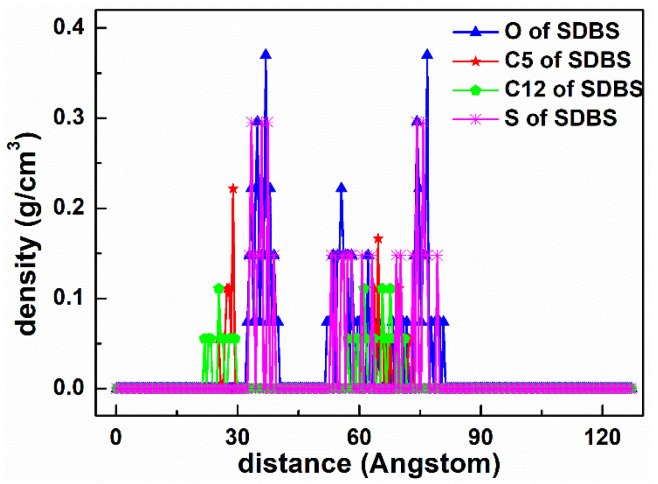
The density distribution profiles of O, C5, C12, and S of SDBS along the z-direction.

**Figure 14 materials-15-03816-f014:**
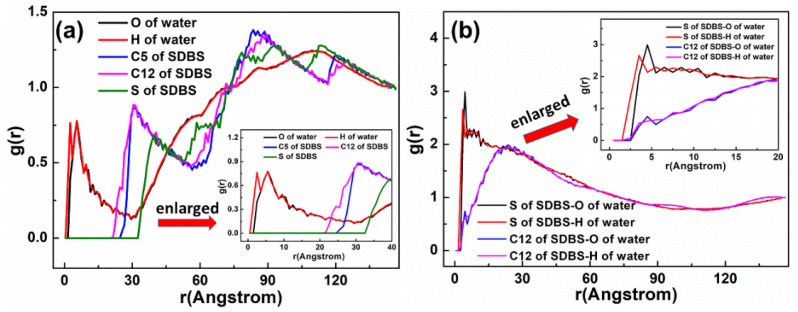
RDF g(r) between the SDBS collector and (**a**) Mcv (001) surface and (**b**) water molecules, respectively.

**Figure 15 materials-15-03816-f015:**
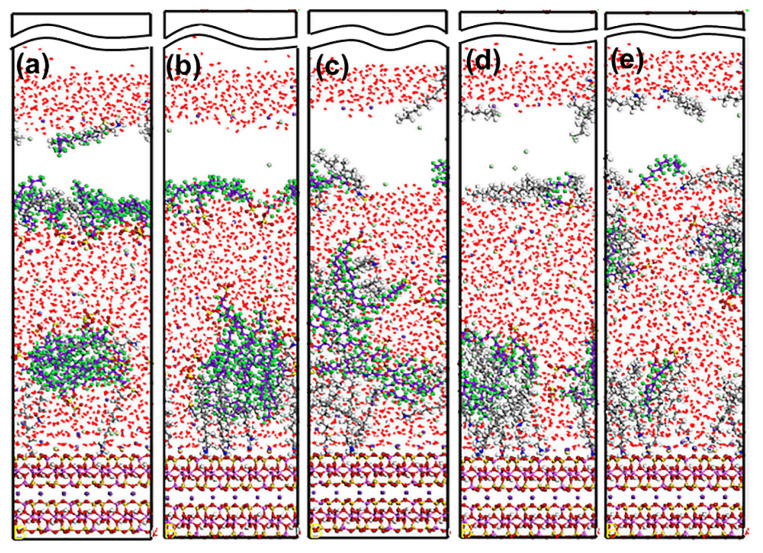
MDS snapshot of mixed DDAHC/SDS collectors forming on the muscovite (001) surface. The molar ratio of DDAHC to SDS is 1:3 (**a**), 1:2 (**b**), 1:1 (**c**), 2:1 (**d**), and 3:1 (**e**). For clarity, only the water molecules are shown in the line model. The different colors represent different atoms, which are the same as in Figure 1 and Figure 2. In addition, the colors light green and light yellow represent chloride and sodium, respectively. The break lines indicate regions of the vacuum slab.

**Figure 16 materials-15-03816-f016:**
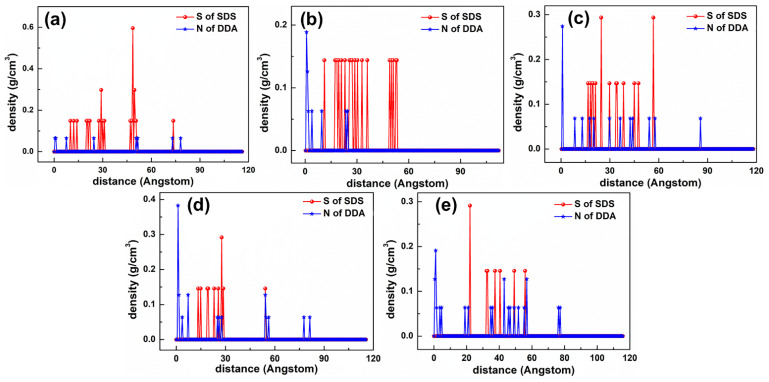
The density distribution profiles of mixed collectors along the z-direction. The molar ratio of DDAHC to SDS is (**a**) 1:3, (**b**) 1:2, (**c**) 1:1, (**d**) 2:1, and (**e**) 3:1.

**Figure 17 materials-15-03816-f017:**
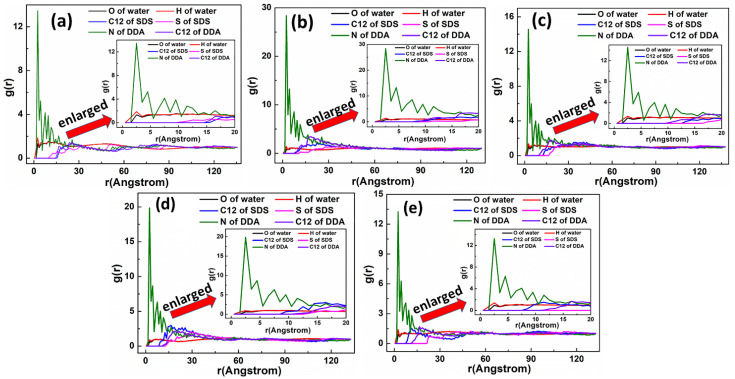
RDF g(r) between the mixed DDAHC/SDS collector and the Mcv (001) surface. The molar ratio of DDAHC to SDS is (**a**) 1:3, (**b**) 1:2, (**c**) 1:1, (**d**) 2:1, and (**e**) 3:1.

**Figure 18 materials-15-03816-f018:**
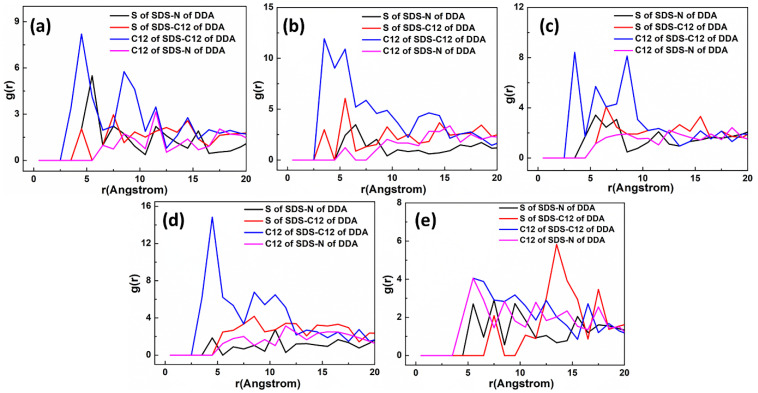
RDF g(r) between the DDAHC and SDS collector. The molar ratio of DDAHC to SDS is (**a**) 1:3, (**b**) 1:2, (**c**) 1:1, (**d**) 2:1, and (**e**) 3:1.

**Figure 19 materials-15-03816-f019:**
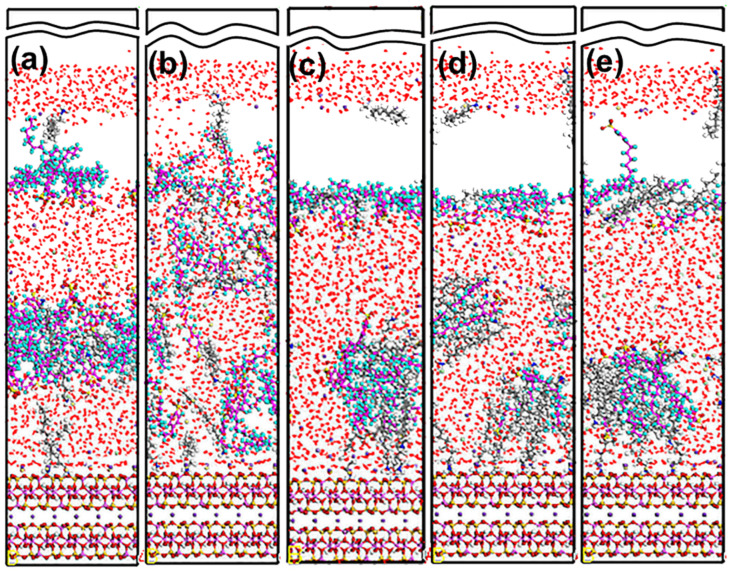
MDS snapshot of mixed DDAHC/SDBS collectors forming on the Mcv (001) surface. The molar ratio of DDAHC to SDS is 1:3 (**a**), 1:2 (**b**), 1:1 (**c**), 2:1 (**d**), and 3:1 (**e**). For clarity, only the water molecules are shown in the line model. The different colors represent different atoms, which are the same as in Figure 1 and Figure 2. In addition, the colors light green and light yellow represent chloride and sodium, respectively. The break lines indicate regions of the vacuum slab.

**Figure 20 materials-15-03816-f020:**
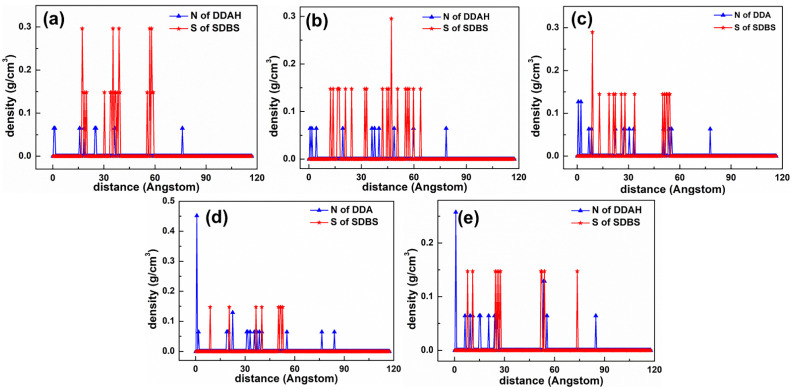
The density distribution profiles of mixed collectors along the z-direction. The molar ratio of DDAHC to SDBS is (**a**) 1:3, (**b**) 1:2, (**c**) 1:1, (**d**) 2:1, and (**e**) 3:1.

**Figure 21 materials-15-03816-f021:**
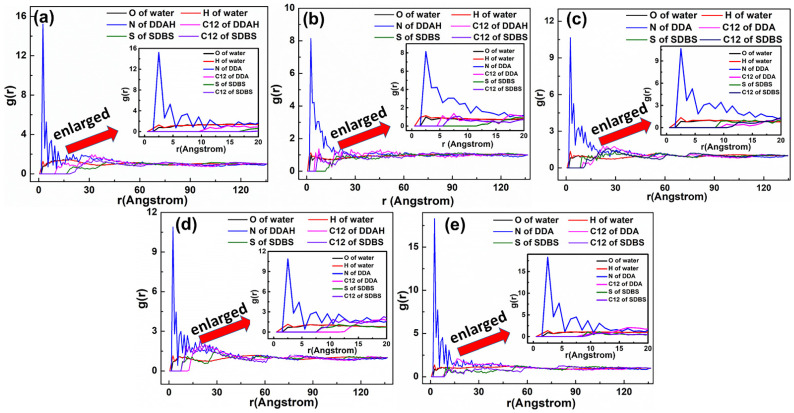
RDF g(r) between the mixed DDAHC/SDBS collector and Mcv (001) surface. The molar ratio of DDAHC to SDBS is (**a**) 1:3, (**b**) 1:2, (**c**) 1:1, (**d**) 2:1, and (**e**) 3:1.

**Figure 22 materials-15-03816-f022:**
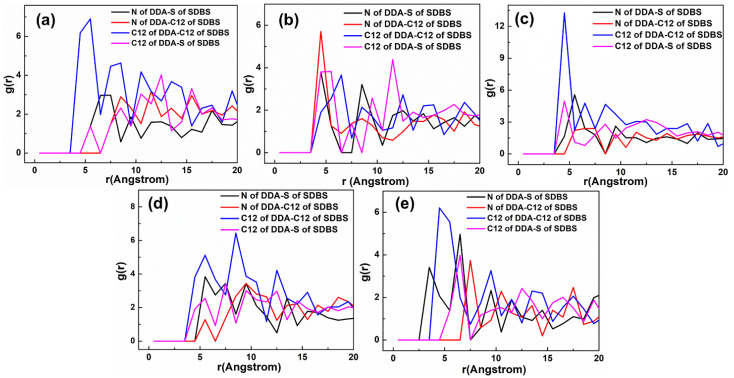
RDF g(r) between the DDAHC and SDBS collector. The molar ratio of DDAHC to SDBS is (**a**) 1:3, (**b**) 1:2, (**c**) 1:1, (**d**) 2:1, and (**e**) 3:1.

**Figure 23 materials-15-03816-f023:**
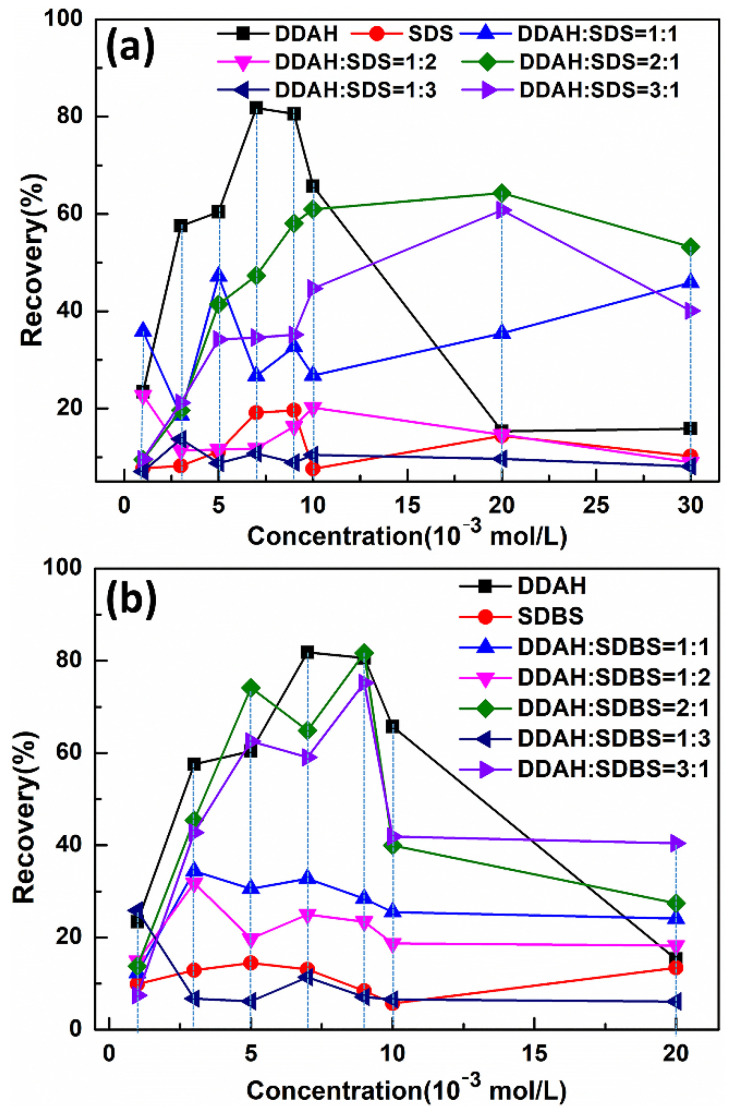
The effect of molar ratio of collectors on flotation recovery of Mcv (**a**) DDAHC+SDS; (**b**) DDAHC + SDBS.

**Table 1 materials-15-03816-t001:** Molar ratios of DDAHC:SDS or DDAHC:SDBS and number of DDA^+^, SDS^−^, SDBS^−^, Cl^−^, Na^+^ in the MDS. For all the systems, the number of K ions and water molecules is set to 30 and 1200, respectively. (D, S, and SD in the table stand for the abbreviations of DDAHC, SDS, and SDBS collectors, respectively.)

Systems	Molar Ratios (D:S or D:SD)	DDA^+^	SDS^−^	SDBS^−^	Cl^−^	Na^+^
D30	-	30	-	-	30	-
S30	-	-	30	-	-	30
SD30	-	-	-	30	-	30
D8S23	1:3	8	23	-	8	23
D10S20	1:2	10	20	-	10	20
D16S15	1:1	16	15	-	16	15
D20S10	2:1	20	10	-	20	10
D23S8	3:1	23	8	-	23	8
D8SD23	1:3	8	-	23	8	23
D10SD20	1:2	10	-	20	10	20
D15SD15	1:1	15	-	15	15	15
D20SD10	2:1	20	-	10	20	10
D23SD8	3:1	23	-	8	23	8

**Table 2 materials-15-03816-t002:** Chemical composition of pure muscovite sample (wt%).

Al_2_O_3_	K_2_O	MgO	SiO_2_	Fe_2_O_3_
31.496	10.843	1.264	46.013	6.802

**Table 3 materials-15-03816-t003:** Interaction energy between the collectors and muscovite surface (kcal/mol). DDAHC, SDS and SDBS S are abbreviations for dodecyl amine hydrochloride, sodium dodecyl sulfate and sodium dodecyl benzene sulfonate, respectively. D, S, and SD in the table stand for the abbreviations of DDAHC, SDS, and SDBS collectors, respectively.

Collectors	System	*E_total_*	*E_mus_*	*E_col_*	*E_inter_*
Single collector	D30	−460,712.9139	−442,048.0052	6881.007	−851.53
	S30	−469,886.5041	−473,503.4898	2132.7143	49.47
	SD30	−468,925.1094	−473,767.5963	3875.312	32.24
Mixed collectors DDAHC + SDS	D8S23	−459,997.1125	−456,393.1465	2752.9275	−205.65
	D10S20	−469,017.6415	−463,698.4841	2380.5639	−256.657
	D16S15	−458,419.7227	−450,551.3448	3032.0479	−351.63
	D20S10	−459,932.9465	−447,300.8663	5073.0874	−590.17
	D23S8	−454,045.5905	−445,243.15	3620.148	−400.73
Mixed collectors DDAHC + SDBS	D8SD23	−463,780.6320	−457,085.4462	2149.4245	−285.31
	D10SD20	−460,565.1272	−450,882.7433	2139.3061	−394.06
	D15SD15	−451,808.8386	−440,744.5416	3442.0143	−483.54
	D20SD10	−455,002.4834	−440,724.5883	5677.6587	−665.19
	D23SD8	−457,303.2624	−445,861.5949	4729.0245	−521.64

**Table 4 materials-15-03816-t004:** The interaction energy values between different collector–mineral systems. (The interaction energy in the literature is calculated according to Equation (2).)

Number	Minerals	Plane	Collectors	ΔE (kcal/mol)
[8]	muscovite	001	DDA^+^	−123.400
[14]	kaolinite	001	10 DDAHC	−277.6000
			10 DDAHC + 5 OA	−183.5333
[20]	kyanite	100	1 Octadecylamine (OA)	−305.5186
			1 SHS	−493.2314
[20]	andalusite	110	1 OA	−253.1644
			1 SHS	−1361.4364
[20]	sillimanite	010	1 OA	−263.0784
[27]	muscovite	001	NaOL	10.3754
			DDA	−27.7817
			DDA-NaOL	−79.2860
[28]	muscovite	001	15 DDA	−1049.10
[40]	siderite	101	10 NaOL	−2350.9587
	hematite	001		−42.2757
	quartz	101		−6259.5622
[41]	spodumene	110	Oleate	−235.80
	anorthite	001		−141.90
	muscovite	001		127.00
[42]	low-rank coal	-	9 dodecyltrimethylammonium bromide (DTAB)	−634.8016
[43]	calcite	-	50 Dodecane (C12)	−215.6400
			50 C12 + 6 sodium hexadecyl sulfonate (SHS)	−180.0714
[44]	magnesite	101	20 cetyl phosphate adsorption (PCP)	−2301.0468
	calcite	104	20 PCP	−162.4522
[45]	magnesite	104	20 PCP	−97.18
	dolomite	104	20 PCP	−22.84

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
