# Peer review of "Molecular Dynamics Simulation Study on the Interactions of Mixed Cationic/Anionic Collectors on Muscovite (001) Surface in Aqueous Solution"

_materials, 2022, doi:10.3390/ma15113816_

Round 1
Reviewer 1 Report
Dear Authors,
The issues described in the manuscripts are still important for mineral processing, especially when it is necessary to minimize the amount of waste generated.
A very detailed analysis of the interaction of muscovite with the collectors used is valuable for the characterization of the surface properties of the mineral. The use of the modeling technique for the evaluation of the type of sorption is very beneficial as it saves time for direct experiments with the use of the trial and error method and enables the indication of the type and number of collectors used in the flotation process. This analysis takes up a significant part of the manuscript and it should be considered whether the entire modeling presented is necessary - please give the authors' opinion on this matter.
I appreciate the determination and accuracy of the authors due to the large number of micro-flotation experiments performed, which were the verification of model findings.
My manuscript, however, requires considerable revision.
General comments and recommendations:
- Introduction: is too short; accumulates quoted literature items without discussing them; one should refer directly to the results of the cited studies to indicate their relationship with this manuscript
- the introduced abbreviations should be used consistently, e.g. MDS (molecular dynamics simulations) or (MD) + simulation
- give the full name of abbreviations used when they appear in the paper for the first time - it is necessary (eg; RDF? line 73); with this in mind, re-edit chapter 2 - not every potential reader is familiar with the modeling system used. The introduced explanations will help him understand the methodology of analysis and research
- used abbreviations should be entered after the full name (description)
- the number of abbreviations used should be verified - are they needed because they repeat, or is it better to remove them because they are unnecessary
- remove unnecessary spaces or insert spaces where they are missing (applies to all the paper)
- the drawings should be placed in the text in a position consistent with the publisher's guidelines (right-hand side or center ???)
- if the drawings / diagrams are generated, for example, by a program used for modeling, this should be indicated in the title of figure
- drawing titles should be written with a capital letter
- the dimensions of the drawings / diagrams / shown in the paper should be max. legible and comparable sizes, and certainly they should be adjusted to the size of the page (especially fig. 15-21) to avoid empty areas - please make the correct editorial
-fig. 22 requires greater readability, consider placing charts - one below the other (a above b); because the values ​​of the yields at a constant concentration of the collector are quoted, in my opinion it is worth plotting vertical lines corresponding to these concentrations in order to make the evaluation of the results easier - especially since the quoted concentrations are not marked on the axis
- variable symbols should be written in italics (mainly chapter 3)- please check the guidelines for authors
-equation 3; if you use this formula for calculations, you will get the result as a percentage ??
-the authors did not provide information about errors for the mean value of the yield, which is analyzed here - this should be completed
Kind Regards,
Reviewer
Reviewer 2 Report
The authors use molecular dynamics to study the interfacial interactions of a mineral adsorbent with three ionic collectors submerged in water.
The authors edit a vacuum zone of 80 A on top of the water layer, presumably because they want to isolate the mineral adsorbent from its periodical images. This setup creates a series of important issues that need to be addressed before the manuscript is considered for publication.
After all, the vacuum zone doesn’t block the periodicity of the mineral adsorbent. It can be seen in figure 5, that some DDAH collectors appear near the top edge of the box, because they interact periodically with the bottom side of the solid. These macromolecules lay in the vacuum zone. The problem is that they are adsorbed by the solid not because they are rejected from the water solution but from a non-interacting environment (i.e., vacuum). In this respect, DDAH adsorption is not equally competitive on the water-side and the vacuum-side interface of the solid.
During the NVT equilibration, the water molecules diffuse into the vacuum zone to yield an appropriate overall density for the water solvent. The water—vacuum interface cannot exist at equilibrium. However, the configurations show that some anionic collectors, SDS and SDBS, remain on the water-vacuum interface.
In figure 3, the authors present fluctuations of the temperature to approve that their system is equilibrated. The temperature is supposed to remain constant because it is an input in NVT, and it is regulated by the Nose Hoover thermostat. To realize that a system has been equilibrated, the authors should display the potential energy and visualize that this has converged.
The authors repeat the lines 276 - 281 in a different section in the manuscript.
What is the length of macromolecules?
I suppose that the authors should better remove the periodicity on the z direction, rather than introducing the vacuum zone.
Reviewer 3 Report
This study aimed to investigate the adsorption behavior of dodecyl amine hydrochloride (DAH) as a cationic collector, sodium dodecyl sulfate (SDS) and sodium dodecyl sulfonate (SDBS) as anionic collectors, and their mixtures at different concentrations/molar ratios on the muscovite surface (001) using Molecular Dynamics Simulations (MDS) in detail.
I see that the authors provided well-designed experimental studies and produced good data, and presented their results in the manuscript very well.
However, some issues must be considered before the manuscript is accepted.
It contains some grammatical problems and must be carefully edited again in terms of grammar and writing.
Besides these, the authors must discuss their results in detail.
And, most importantly, it must be shown differences in their results compared to the data with more references from the literature.
In addition to them, please see the attached file where you can find my corrections/comments, and respond to them carefully one by one.
Overall, the manuscript can be accepted after the correction/revision, then it should be considered to be published in the Journal of Materials.

Round 2
Reviewer 2 Report
About the vacuum zone (partition) and the periodicity on the z direction.
The authors say that if the vacuum gap is small (or removed) the bottom surface of the crystal will influence the top of the system. There will be no such an influence if periodicity is removed on the z direction.
It should be clear that the use of a vacuum partition is wrong, even though other researchers may have used it. Firstly, the water molecules should expand into the vacuum partition at equilibrium. Notably, the expansion of water is not shown in the revised manuscript. Secondly, because water is polar whereas vacuum is non-interacting. The collectors that reside inside the water partition are adsorbed on the surface because they are rejected by the water solution. The collectors that reside in vacuum, they are rejected by what?
The authors should present some more references of molecular simulation studies in which the simulation boxes are designed appropriately.
One example can be doi.org/10.3390/app12073460, where MD simulations are used to visualize how graphene oxide layers may exfoliate from a support film in water solution.
Another example can be doi.org/10.3390/molecules27030956, where the box is designed implicitly to simulate the interface of two solvents (one is water, the other is hydrophobic), that is crossed by different carbon nanoparticles.
